# Prevalence of opportunistic bacterial infections (tuberculosis and pneumonia) among people with HIV in Ethiopia: Systematic review and meta-analysis

Aleka Aemiro[1], Abayeneh Girma[1]*, Getachew Alamnie[1], Demsew Beletew[2], Amere Genet[1]

**1** Department of Biology, College of Natural and Computational Sciences, Mekdela Amba University, Tulu Awuliya, Ethiopia, **2** Department of Statistics, College of Natural and Computational Sciences, Mekdela Amba University, Tulu Awuliya, Ethiopia

* gabayeneh2013@gmail.com

## Abstract

### Background

Individuals with weakened immune systems, such as those living with Human Immunodeficiency Virus (HIV), are more vulnerable to opportunistic bacterial infections, which include tuberculosis and pneumonia. This systematic review and meta-analysis looked at the pooled prevalence of opportunistic bacterial infections among people living with HIV in different regions of Ethiopia.

### Methods

By looking through open online databases, articles written in English were considered. Joanna Briggs Institute's critical appraisal tool for prevalence study was used to check the quality of each article. Inverse variance ($I^2$), sensitivity analysis, funnel plot, and Egger's regression tests were used to check heterogeneity and publication bias. Because of a high heterogeneity, a random-effects model was used to estimate the pooled prevalence of opportunistic bacterial infections among people living with HIV.

### Results

About 18.06% (1824/9651) with (95% CI: 14.09–22.02) of the pooled population had opportunistic tuberculosis from 20 studies included, while from 16 included studies, the pneumonia infection was 11.64% (1040/8095/) with (95% CI: 8.45–14.83).

### Conclusion

The prevalence of tuberculosis and pneumonia among people living with HIV in Ethiopia is high. Therefore, policymakers and health planners should put a great deal of emphasis on the implementation of relevant prevention and control measures.

**Data availability statement:** All relevant data are included in the paper and its Supporting information files.

**Funding:** The author(s) received no specific funding for this work.

**Competing interests:** The authors have declared that no competing interests exist.

**Abbreviations:** AOR, Adjusted Odds Ratio; CI, Confidence Interval; GRADE, Grading of Recommendations Assessment, Development and Evaluation; OIs, Opportunistic infections; PLHIV, People Living with Human Immunodeficiency Virus; PRISMA, Preferred Reporting Items for Systematic reviews and Meta-Analyses; SNNPR, Southern Nations, Nationalities and People's Region; SSA, Sub-Saharan Africa.

## Registration

The review was registered in the International Prospective Register of Systematic Reviews (PROSPERO) with the registration number "CRD42024587645", on September 17, 2024.

## Introduction

Opportunistic infections (OIs) are caused by microorganisms, such as bacteria, fungi, parasites and viruses that typically do not cause disease in healthy people but can become pathogenic when the host's immune system is weakened [1]. Over 80% of AIDS (Acquired immunodeficiency syndrome)-related deaths are caused by OIs, which are intestinal parasites and bacteria that more easily infect immunocompromised patients [2]. In 2023, there were 39.9 million people living with HIV globally, and 630,000 deaths from HIV-related causes [3], including 1.4 million children (under the age of 15) [4]. A similar year report also states that OI was the main cause of nearly 76,000 million child deaths from HIV-related causes [5].

Children living with HIV and newborns who have been exposed to it are more likely to suffer from bacterial respiratory tract infections than HIV-uninfected children [6]. Tuberculosis (TB) and pneumonia are the most frequent causes of hospital admissions among all HIV-related OIs for children living with HIV on antiretroviral therapy (ART) in Europe, East China, and other low- and middle-income countries [7]. In 2017, an estimated 6.3 million fatalities were reported worldwide , with children between the ages of 5 and 14 accounting for one million of those deaths [8]. In 2022, TB was the second-leading cause of death from an infectious disease worldwide, with COVID-19 being the first. Additionally, while 10.6 million people fell ill with TB in 2022, 7.5 million were newly diagnosed [9]. In children worldwide, pneumonia is the most common cause of sickness and death [10].

In Ethiopia, pneumonia is a major health threat to children under five, where it is a leading cause of death, accounting for about 18% of all deaths in this age group [11]. Reports from a 2020 meta-analysis of 12 studies indicate the pooled prevalence of pneumonia among children under five in Ethiopia was 20.68% [12]. Globally, Ethiopia ranks sixth among the top 15 countries in terms of morbidity and mortality from pneumonia [13] and the burden is high with pneumonia/HIV patients. Regarding TB, in 2022, around 151,000 people in Ethiopia were diagnosed for TB, leading to over 19,000 deaths each year due to the illness. Moreover, Ethiopia is one of the 30 countries with a high TB and TB/HIV burden; with an estimated annual TB incidence of 126/100,000 persons and a death rate of 12 per 100,000 people, according to the 2023 Global TB Report [9].

Previously, Woldegeorgis and colleagues [14] conducted a systematic review and meta-analysis regarding the prevalence and determinants of OIs among HIV-infected adults receiving ART in Ethiopia. However, the prevalence of opportunistic bacterial infections, particularly tuberculosis and pneumonia, among people of all age groups living with HIV in the country is not collected, well organised, and documented as a

systematic review and meta-analysis. Therefore, the objective of this systematic review and meta-analysis is to estimate the overall prevalence of opportunistic bacterial infections among people living with HIV from available research conducted in different regions of Ethiopia. The findings of this work will be used by the concerned stakeholders to reduce the prevalence of opportunistic bacterial infections and design evidence-based interventions.

## Methods

### Design and protocol registration

This systematic review and meta-analysis was designed to estimate the country pooled prevalence of opportunistic bacterial infections among people with HIV in Ethiopia. The result was reported following Preferred Reporting Items for Systematic Reviews and Meta-Analysis (PRISMA)-2020 guideline [15] (see S1 Table). The review protocol was registered in the International Prospective Register of Systematic Reviews (PROSPERO) under registration number CRD42024587645.

### Search strategy

Electronic databases such as, ScienceDirect, PubMed, and the Cochrane Library were used in a comprehensive search (S2 Table). Core search terms and phrases such as "Prevalence," "epidemiology," "magnitude," "opportunistic infections," "opportunistic bacterial infections," "people with HIV," and "Ethiopia" were used to search articles. The work was limited to studies with clearly stated sample sizes, numbers of positive cases, and study locations. The study was conducted between September 06/2024 and December 1/2024 and the study's objective was to determine the prevalence of opportunistic bacterial infections among people with HIV in Ethiopia.

### Eligibility criteria of the studies

The articles collected through the searches were evaluated for inclusion in the meta-analysis based on the following criteria: (i) only prospective and retrospective studies on opportunistic bacterial infections, (ii) reports published only in English, (iii) studies that reported prevalence (iv), studies conducted in Ethiopia (v) and articles published between 2013 and 2023. Excluded from this review were studies with no full-text access, publications that were duplicated or expanded upon from an original study, articles with insufficient information, research results derived from subjective opinions, articles reported outside the parameters of the outcome of interest, case reports, unpublished data, qualitative investigation design, abstracts, conference presentations and previous systematic reviews.

### Study selection process

Articles found through the electronic database search were entered into an EndNote X9 reference program, where duplicate studies were removed. Two authors (AA and AG) independently screened the titles and abstracts obtained from the search according to the inclusion criteria. To assess if multiple authors' systematic reviews of the same topic are similar, inter-rater agreement (IRA) is calculated using Cohen's kappa, often with the Cochrane Handbook as a guide for methods and interpretation of agreement [16]. The screened articles were then reviewed for a full-text assessment by two independent authors (GA and DB), to ensure that only eligible studies are included in the final review and to minimize bias. Predefined eligibility criteria were used to determine which records were irrelevant and which should be included in the review. If additional information was needed to answer eligibility questions, other authors were involved. Disagreements were resolved through discussion. In addition, reasons for excluding articles were recorded at each stage.

### Data extraction

Two authors (AA and AG) independently abstracted the relevant data using a standardized Microsoft excel 2016 spreadsheet. The data extraction process includes the following details: The first author's name, the year of publication, the

regions of the country, the study design, the sample size, the number of positive cases, and the prevalence of opportunistic bacterial infections. If the study was conducted over an extended period of time in a single field, the most recent year falling within the designated range was used.

### Quality assessment of individual studies

The articles that meet the eligibility criteria and are sufficiently valid for our research question underwent full-text appraisal. The Joanna Briggs Institute (JBI) critical checklist for prevalence studies was used for quality appraisal using 9 criteria [17]. For each question, a score was assigned (no for 'not reported or not appropriate' and yes 'for reported'); the scores were summarized across the items to achieve a total score of 0–9, for prevalence studies. The studies were then classified as low (≤ 2), medium (3–4), and high (≥ 5) quality based on the points awarded. Studies with a final quality score of 50% or higher were considered for inclusion in the systematic review and meta-analysis (S1 and S2 Tables).

### Data analysis

Because of a high heterogeneity, a random-effects model was used to determine the pooled effect sizes. To uncover sources of heterogeneity, we conduct a sensitivity analysis (the impact of each individual studies on the pooled effect size) and subgroup analysis based on the study's region, sample size, and year of publication. The inverse variance ($I^2$) statistics were used to assess the magnitude of heterogeneity of the included articles. The $I^2$ test was used to assess the heterogeneity across studies with interpretations assigned to $I^2$ values: 0% (no heterogeneity), 0–25% (low heterogeneity), 25–50% (medium heterogeneity), and >75% (high heterogeneity) [18]. The risks of publication bias for studies have been examined using funnel plot symmetry and the Egger's test. During Egger's test, p-values less than 0.05 were an indication of a significant presence of publication bias [19]. This meta-analysis was conducted using Stata version 14 software (StataCorp LLC 4905 Lakeway Drive College Station, Texas 77845−4512, USA) with "metan" command.

## Results

A total of 276 articles were retrieved on the prevalence of OIs among people living with HIV in Ethiopia. One hundred twenty five of these articles were excluded due to duplicates. Of the remaining 141 articles, 114 were excluded based on specific criteria included in the inclusion criteria and data extraction protocol. Of the remaining 27 articles, 7 articles were further removed due to lack number of positive cases. Therefore, 20 of the studies met the eligibility criteria and were included in the final systematic review and meta-analysis (Fig 1).

### Characteristics of the eligible studies

Table 1 represents the features of the studies that were subjected to this meta-analysis, concerning the pooled prevalence of TB among people living with HIV. The meta-analysis had twenty (20) studies that matched the eligibility criteria. All studies were cross-sectional carried out between 2013 and 2023. Regarding the study year, seven, eight and five studies were conducted between 2013 and 2018, 2019 and 2021, and 2022 and 2023, respectively. Based on the criteria, Addis Ababa (2 articles), Amhara (9 articles), Oromia (4 articles), SNNPR (4 articles), and Tigray (1 article) were involved. The prevalence of opportunistic TB infections among people living with HIV ranged from 7.1% to 43.5% (Table 1).

Table 2 lists the features of the studies that were subjected to a meta-analysis, concerning the pooled prevalence of pneumonia among people living with HIV. The meta-analysis of pneumonia prevalence includes sixteen (16) studies that matched the eligibility criteria. All studies were cross-sectional carried out between 2013 and 2023. Three, seven, and six investigations on the prevalence of pneumonia were conducted between 2013 and 2015, 2018 and 2020, and 2021 and 2023, respectively. Addis Ababa (2 studies), Amhara (7 articles), Oromia (4 articles), and SNNPR (3 articles) regions

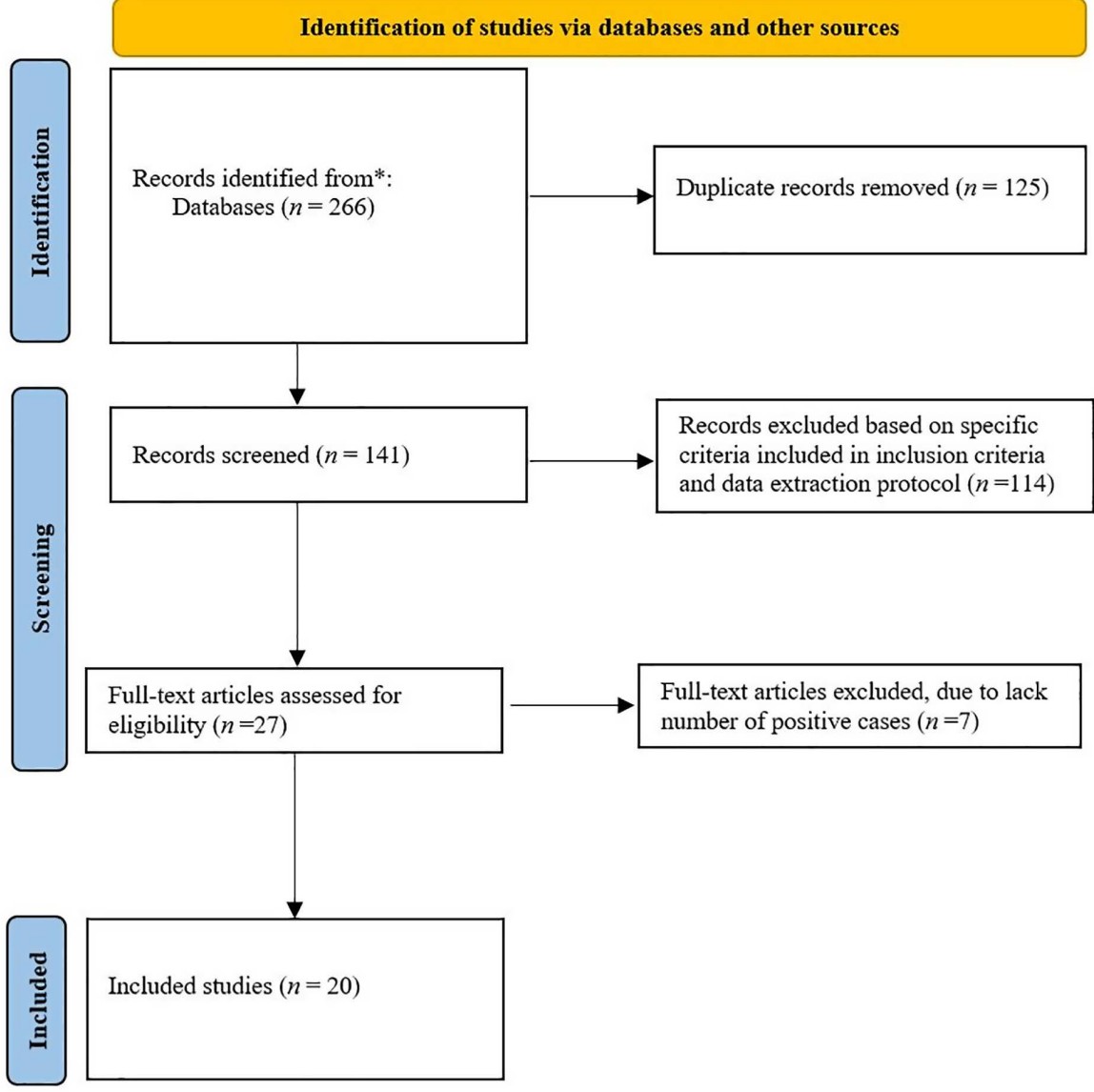

**Fig 1. Flow diagram summarizing the selection of eligible studies.**

were taken into consideration in this meta-analysis of pneumonia prevalence. Table 2 shows that among studies that were eligible, the prevalence of opportunistic pneumonia infections varied from 1% to 29.8%.

## Pooled prevalence of opportunistic bacterial infections

The pooled prevalence of tuberculosis and pneumonia infection among people living with HIV in Ethiopia was estimated using a random-effects model because of the high heterogeneity among studies. The pooled prevalence of tuberculosis infection among people living with HIV in Ethiopia was 18.06% (95% CI: 14.09−22.02) with observed heterogeneity ($I^2$ = 95.9, p < 0.001) (Fig 2) and pneumonia infection was 11.64% (95% CI: 8.45−14.83) with observed heterogeneity ($I^2$ = 95.5, p < 0.001) (Fig 3).

**Table 1. Characteristics of included studies to calculate the pooled prevalence of TB among people living with HIV.**

| Author | Publication year | Region | Study design | Sample size | Case | Prevalence (%) |
|--------|------------------|--------|--------------|-------------|------|----------------|
| Damtie *et al.* | 2013 [20] | Amhara | Cross-sectional | 360 | 35 | 9.72 |
| Moges & Kassa | 2014 [21] | Amhara | Cross-sectional | 423 | 41 | 9.7 |
| Mitku *et al.* | 2015 [22] | Oromia | Cross-sectional | 358 | 76 | 21.23 |
| Alemayehu *et al.* | 2017 [23] | SNNPR | Cross-sectional | 362 | 70 | 19.4 |
| Deribe & Estifanos | 2018 [24] | A/Ababa | Cross-sectional | 315 | 137 | 43.5 |
| Solomon *et al.* | 2018 [25] | SNNPR | Cross-sectional | 744 | 133 | 18.0 |
| Urgessa *et al.* | 2018 [26] | Oromia | Cross-sectional | 418 | 55 | 13.2 |
| Dereje *et al.* | 2019 [27] | A/Ababa | Cross-sectional | 384 | 38 | 9.8 |
| Melkamu *et al.* | 2020 [28] | Amhara | Cross-sectional | 408 | 122 | 29.8 |
| Fite & Aga | 2020 [29] | Oromia | Cross-sectional | 497 | 78 | 15.7 |
| Wachamo & Bonja | 2020 [30] | SNNPR | Cross-sectional | 420 | 48 | 11.6 |
| Weldearegawi *et al.* | 2020 [31] | Tigray | Cross-sectional | 400 | 38 | 9.5 |
| Tegegne *et al.* | 2020 [32] | Amhara | Cross-sectional | 354 | 75 | 21.1 |
| Chanie *et al.* | 2021 [33] | Amhara | Cross-sectional | 349 | 25 | 7.1 |
| Dembelu & Wosenelh | 2021 [34] | SNNPR | Cross-sectional | 450 | 94 | 21.0 |
| Kebede *et al.* | 2022 [35] | Amhara | Cross-sectional | 405 | 111 | 27.5 |
| Mequanente *et al.* | 2022 [36] | Amhara | Cross-sectional | 389 | 41 | 10.6 |
| Birmeka | 2023 [37] | Oromia | Cross-sectional | 1448 | 408 | 28.2 |
| Dagnaw *et al.* | 2023 [38] | Amhara | Cross-sectional | 715 | 78 | 10.9 |
| Mekonnen *et al.* | 2023 [39] | Amhara | Cross-sectional | 452 | 121 | 26.7 |

**Table 2. Characteristics of included studies to calculate the pooled prevalence of pneumonia among people living with HIV.**

| Author | Publication year | Region | Study design | Sample size | Case | Prevalence (%) |
|--------|------------------|--------|--------------|-------------|------|----------------|
| Damtie *et al.* | 2013 [20] | Amhara | Cross-sectional | 360 | 5 | 1.38 |
| Moges & Kassa | 2014 [21] | Amhara | Cross-sectional | 423 | 25 | 5.9 |
| Mitku *et al.* | 2015 [22] | Oromia | Cross-sectional | 358 | 10 | 2.8 |
| Deribe & Estifanos | 2018 [24] | A/Ababa | Cross-sectional | 315 | 38 | 12.1 |
| Solomon *et al.* | 2018 [25] | SNNPR | Cross-sectional | 744 | 121 | 16.3 |
| Urgessa *et al.* | 2018 [26] | Oromia | Cross-sectional | 418 | 42 | 10.1 |
| Dereje *et al.* | 2019 [27] | A/Ababa | Cross-sectional | 384 | 4 | 1.0 |
| Melkamu *et al.* | 2020 [28] | Amhara | Cross-sectional | 408 | 122 | 29.8 |
| Fite & Aga | 2020 [29] | Oromia | Cross-sectional | 497 | 29 | 5.8 |
| Wachamo & Bonja | 2020 [30] | SNNPR | Cross-sectional | 420 | 90 | 21.5 |
| Dembelu& Wosenelh | 2021 [34] | SNNPR | Cross-sectional | 450 | 45 | 10.0 |
| Chanie *et al.* | 2021 [33] | Amhara | Cross-sectional | 349 | 13 | 3.7 |
| Tegegne *et al.* | 2022 [32] | Amhara | Cross-sectional | 354 | 53 | 14.9 |
| Birmeka | 2023 [37] | Oromia | Cross-sectional | 1448 | 166 | 11.5 |
| Dagnaw *et al.* | 2023 [38] | Amhara | Cross-sectional | 715 | 149 | 20.8 |
| Mekonnen *et al.* | 2023 [39] | Amhara | Cross-sectional | 452 | 128 | 28.3 |

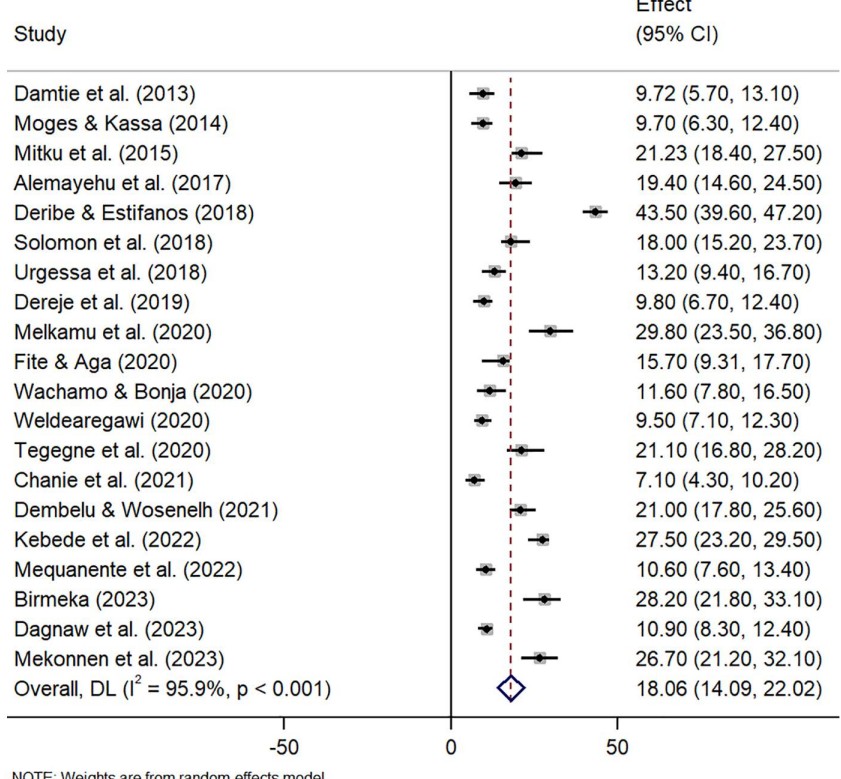

**Fig 2. Overall pooled prevalence of tuberculosis among HIV positive patients in Ethiopia.**

## Sub-group analysis of tuberculosis among people living with HIV

Subgroup analysis was performed using sample size, study region, and year of publication. When we compare the pooled prevalence of tuberculosis, the prevalence is higher in studies with a sample size of 384 and below (18.79%; 95% CI: 9.02, 28.56) than in studies with a sample size greater than 384 (17.57%; 95% CI: 13.71, 21.42) (Table 3).

The Addis Ababa area of Ethiopia had the greatest pooled prevalence of tuberculosis infections among people living with HIV, reported at 26.63% (95% CI: −6.40−59.65), while Tigray had the lowest prevalence, recorded at 9.5% (95% CI: 6.9–12.1) (Table 3).

The greatest recorded pooled prevalence of tuberculosis during the research period was 20.57% (95% CI: 12.35–28.80) between 2022 and 2023, followed by 19.23%; 95% CI: 10.13−28.34) between 2013 and 2018, and the lowest was between 2019 and 2021 (15.22%; 95% CI: 10.83−19.61) (Table 3), with high heterogeneity in all groups.

## Sub-group analysis of pneumonia among people living with HIV

Subgroup analysis was performed using sample size, study region, and year of publication. The pooled prevalence of pneumonia is lower in studies with a sample size of 384 and below (5.58%; 95% CI: 2.24, 8.92) compared with those with a sample size greater than 384 (15.53%; 95% CI: 10.97, 20.09) (Table 4), with high heterogeneity in both groups.

Regarding study regions, Addis Ababa had the lowest prevalence of pneumonia infections at 6.47% (95% CI: −4.41−17.35), the SNNPR region of Ethiopia had the highest pooled prevalence among people living with HIV, at 15.65% (95% CI: 9.31−21.99) (Table 4), with high heterogeneity in both groups.

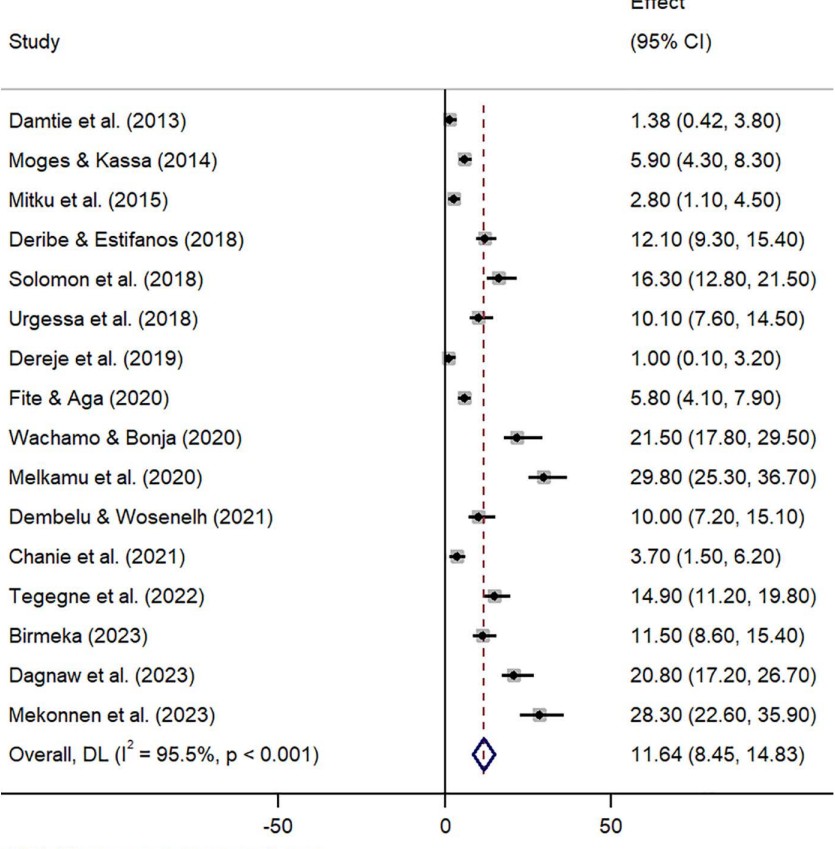

Study | Effect (95% CI)

| Study | Effect (95% CI) |
|---|---|
| Damtie et al. (2013) | 1.38 (0.42, 3.80) |
| Moges & Kassa (2014) | 5.90 (4.30, 8.30) |
| Mitku et al. (2015) | 2.80 (1.10, 4.50) |
| Deribe & Estifanos (2018) | 12.10 (9.30, 15.40) |
| Solomon et al. (2018) | 16.30 (12.80, 21.50) |
| Urgessa et al. (2018) | 10.10 (7.60, 14.50) |
| Dereje et al. (2019) | 1.00 (0.10, 3.20) |
| Fite & Aga (2020) | 5.80 (4.10, 7.90) |
| Wachamo & Bonja (2020) | 21.50 (17.80, 29.50) |
| Melkamu et al. (2020) | 29.80 (25.30, 36.70) |
| Dembelu & Wosenelh (2021) | 10.00 (7.20, 15.10) |
| Chanie et al. (2021) | 3.70 (1.50, 6.20) |
| Tegegne et al. (2022) | 14.90 (11.20, 19.80) |
| Birmeka (2023) | 11.50 (8.60, 15.40) |
| Dagnaw et al. (2023) | 20.80 (17.20, 26.70) |
| Mekonnen et al. (2023) | 28.30 (22.60, 35.90) |
| Overall, DL ($I^2$ = 95.5%, p < 0.001) | 11.64 (8.45, 14.83) |

NOTE: Weights are from random-effects model

**Fig 3. Overall pooled prevalence of pneumonia among HIV positive patients in Ethiopia.**

**Table 3. Subgroup analysis of the magnitude of tuberculosis among people with HIV in Ethiopia.**

| Variables | Characteristics | Included studies | Sample size | Prevalence (95% CI) | $I^2$, p–value |
|---|---|---|---|---|---|
| Sample size | ≤384 | 7 | 2482 | 18.79 (95% CI: 9.02, 28.56) | 97.8, p < 0.001 |
| | >384 | 13 | 7169 | 17.57 (95% CI: 13.71, 21.42) | 93.5, p < 0.001 |
| Region | Oromia | 4 | 2721 | 19.32 (95% CI: 13.29, 25.36) | 86.5, p < 0.001 |
| | SNNPR | 4 | 1976 | 17.52 (95% CI: 13.4, 21.63) | 72.1, p < 0.013 |
| | Amhara | 9 | 3855 | 16.68 (95% CI: 11.50, 21.87) | 95.2, p < 0.001 |
| | Tigray | 1 | 400 | 9.5 (95% CI: 6.9, 12.1) | 0.0,_______ |
| | Addis Ababa | 2 | 699 | 26.63 (95% CI: −6.40, 59.65) | 99.5, p < 0.001 |
| Publication Year | 2013–2018 | 7 | 2980 | 19.23(95% CI: 10.13, 28.34) | 97.4, p < 0.001 |
| | 2019–2021 | 8 | 3262 | 15.22 (95% CI: 10.83, 19.61) | 90.9, p < 0.001 |
| | 2022-2023 | 5 | 3409 | 20.57 (95% CI: 12.35, 28.80) | 96.7, p < 0.001 |
| | **Overall** | **20** | **9651** | **18.06 (95% CI: 14.09, 22.02)** | **95.9, p < 0.001** |

*SNNPR*, Southern Nations, Nationalities and People's Region

Table 4. Subgroup analysis of the magnitude of pneumonia among people with HIV in Ethiopia.

| Variables | Characteristics | Included studies | Sample size | Prevalence (95% CI) | $I^2$, P–value |
|---|---|---|---|---|---|
| Sample size | ≤384 | 6 | 2120 | 5.58 (95% CI: 2.24, 8.92) | 93.2, P<0.001 |
| | >384 | 10 | 5975 | 15.53 (95% CI: 10.97, 20.09) | 94.1, P<0.001 |
| Region | Oromia | 4 | 2721 | 7.30 (95% CI: 3.57, 11.03) | 89.5, P<0.001 |
| | SNNPR | 3 | 1614 | 15.65 (95% CI: 9.31–21.99) | 82.0, P=0.004 |
| | Amhara | 7 | 3061 | 14.46 (95% CI: 8.00, 20.92) | 96.9, P<0.001 |
| | Addis Ababa | 2 | 699 | 6.47 (95% CI: −4.41, 17.35) | 97.5, P<0.001 |
| Publication Year | 2013-2015 | 5 | 1141 | 3.31 (95% CI: 0.82, 5.8) | 82.8, P<0.003 |
| | 2018-2020 | 8 | 3186 | 13.4 (95%CI: 7.44, 19.36) | 96.6, P<0.001 |
| | 2021-2023 | 5 | 3768 | 14.50 (95% CI: 8.13, 20.88) | 93.9, P<0.001 |
| | *Overall* | **16** | **8095** | **11.64 (95% CI: 8.45, 14.83)** | **95.5, P<0.001** |

*SNNPR*, Southern Nations, Nationalities and People's Region

The highest pooled prevalence of pneumonia among people with HIV was recorded during the 2021–2023 study period at 14.50% (95% CI: 8.13−20.88). This was followed by the 2018–2020 period at 13.4% (95%CI: 7.44−19.36), and the lowest prevalence was recorded between 2013 and 2015 at 3.31% (95% CI: 0.82−5.8) (Table 4). High heterogeneity was observed in all groups.

## Quality assessment and publication bias

Information on the quality assessment of individual studies is presented in S1 and S2 Tables. Briefly, 100% of the included studies were of high quality (low risk of bias). The asymmetry of the funnel plot indicated the existence of publication bias among the included studies (Figs 4–5). Similarly, the Egger test (Figs 6–7) revealed statistically significant publication bias (P=0.001). Because of a high level of heterogeneity in the included studies, a sensitivity analysis was performed by removing single study one at a time to assess the impact of each study on the pooled effect size. During the sensitivity analysis, studies not included in Figs 8–9 had relatively determinant effects on the overall magnitude of tuberculosis and pneumonia among people living with HIV in Ethiopia.

## Discussion

This systematic review and meta-analysis aimed to study the pooled prevalence of opportunistic bacterial infections among people living with HIV in Ethiopia. Tuberculosis (TB) is the most prevalent and dangerous opportunistic disease among people living with HIV, serving as the clinical manifestation of AIDS in more than half of cases in developing countries. TB was declared a worldwide emergency in 1993. More recently, the Director General of the WHO proclaimed AIDS to be an emergency as well. Pneumonia, which is an inflammation of the lung parenchymal structure caused by aspiration, inhalation, or hematogenous-spread pathogens, is also one of the most common opportunistic infections, particularly in children living with HIV.

The overall national prevalence of tuberculosis infection was 18.06%. The result is almost in accordance with the result of another meta-analysis done in Ethiopia [14]. The finding is also in agreement with results from India, which reported a TB/HIV co-infection rate of 17% [40]; South Africa (17.3%) [41]; Nigeria (16.8%) [42,43]; and Rwanda (20%) [44]. Whereas, a higher prevalence has been reported in Iran (38.2%) [45], Nepal (27.3%), and Nigeria (22.7%) [46,47]. On the other hand, the pooled prevalence was lower than a study conducted in Tanzania (8.3%) [48], Mozambique (10.1%) [49], another study from Tanzania (7.9%) [50], USA (11.5%) [51] and China (7.2%) [52]. This difference might be due to variations in sample size, study design, and diagnosis method. Moreover, the discrepancy could stem from differences

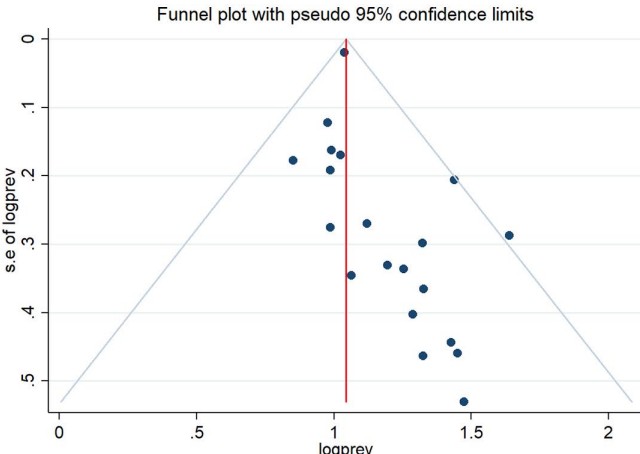

**Fig 4. Funnel plot representing evidence of publication bias in tuberculosis studies among people living with HIV.**

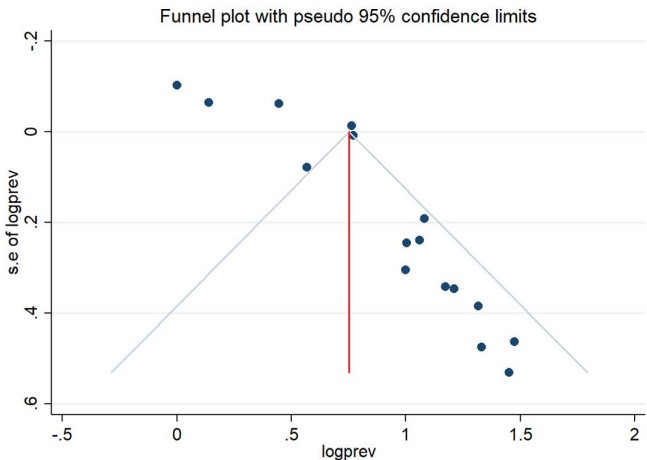

**Fig 5. Funnel plot representing evidence of publication bias in pneumonia studies among people living with HIV.**

in geographical areas of study participants, high exposure to infectious agents, social-economic status, drug resistance, immunity, and nutrition. All these factors may affect the magnitude of tuberculosis, as well as the community's awareness to seek healthcare for TB and HIV.

The overall national prevalence of pneumonia infection was 11.64%. The pooled prevalence found in this current review is in line with a previous report conducted in Ethiopia, which found a pooled prevalence of 12.50% [14]. Pneumonia prevalence in patients was 31.2% [45]. However, our finding is lower than those reported in several other African nations, including Uganda (25.6%) [53], Nigeria (31.6%) [54], Sudan (65%) [55], Kenya (74%) [56] and a previous systematic review and meta-analysis conducted in Iran 31.2% [45]. Conversely, our finding is higher than results from Vietnam (5%) [57], India (5%) [58], Mali (6.7%) [59], South Africa (0.5%) [60] and Malawi (1.0%) [61]. Variations among studies could arise from varied settings, environmental-related factors, household facilities, and the sociodemographic characteristics of mothers or caregivers.

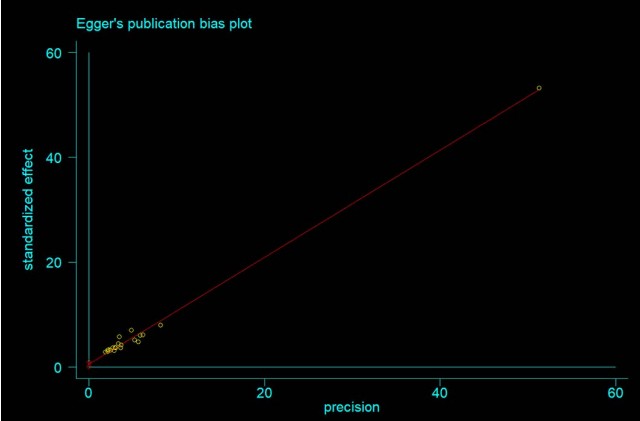

**Fig 6. Egger's publication bias plot of in tuberculosis studies among people living with HIV.**

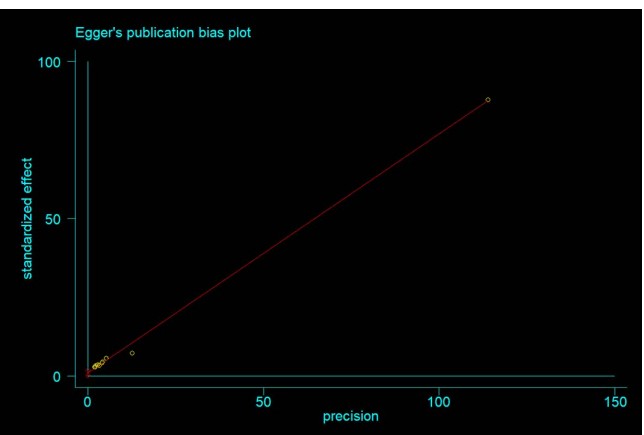

**Fig 7. Egger's publication bias plot of in pneumonia studies among people living with HIV.**

The highest prevalence of tuberculosis among people living with HIV in Ethiopia was reported from Addis Ababa (26.63%). The finding of this result is comparable to studies conducted in the Jimma zone (28.1%) [62], Afar (26.4%) [63] and Shashamene (24.9%) [64]. In addition, it is consistent with the study done in Nigeria (24.5%) [65] and Albania (27.4%) [66]. In contrast, the pooled prevalence in this review for the Tigray region is accounting 9.5%. The variations in different regions might be due to the environmental, socioeconomic, and educational status of study participants.

The highest pooled prevalence of pneumonia is reported from SNNPR region (15.65%). The result is lower than the findings from Goncha Siso Enesie (24.3%), Sheka zone (23.8%) [67], Gondar (26.3%) [68], and Uganda (25.6%) [53]. Nevertheless, the lowest prevalence of pneumonia infections was observed in Addis Ababa (6.47%), which is in harmony with studies conducted in Uganda (6.9%) [69] and Kenya (6.9%) [70]. The difference in prevalence of pneumonia across different regions of Ethiopia could be due to variations in socio-demographic factors (such as race/ethnicity and immigration status), geographical inequality, seasonal variation, socioeconomic differences, and environmental change.

The highest pooled prevalence of tuberculosis was recorded between 2022 and 2023 at 20.57%, followed by 2013 and 2018 at 19.23%, whereas the lowest prevalence was recorded between 2019 and 2021 at 15.22%. Regarding pneumonia,

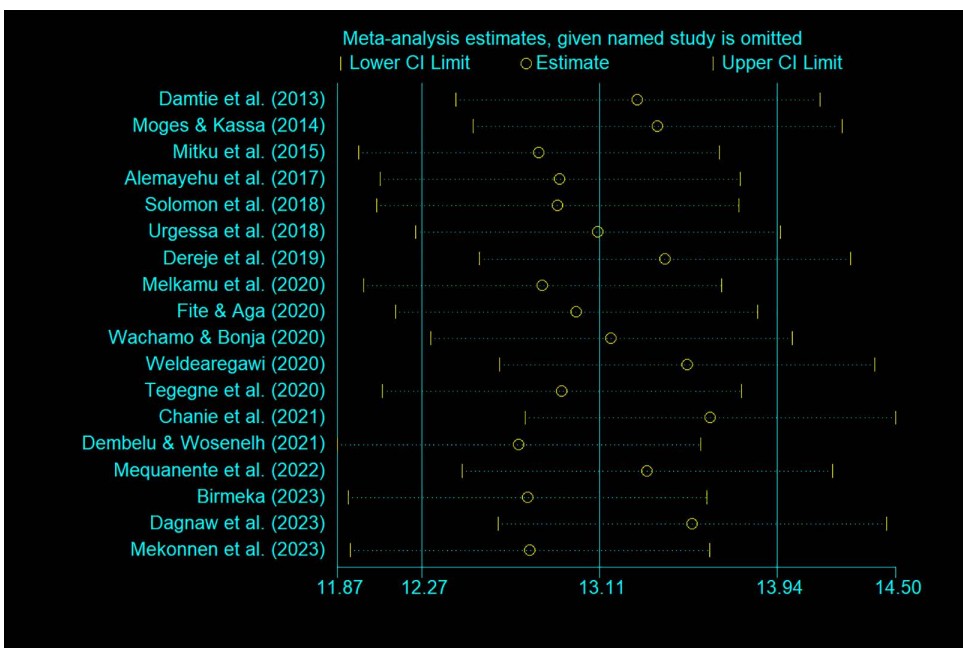

**Fig 8. Sensitivity analysis result of the included studies that assessed the impact of each study on the overall magnitude of tuberculosis among people living with HIV.**

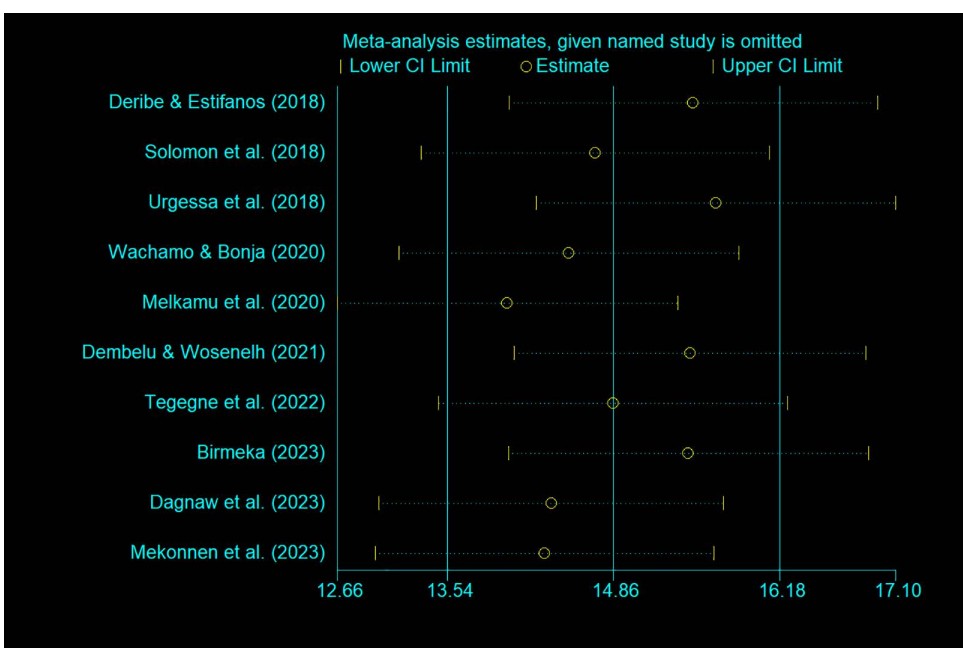

**Fig 9. Sensitivity analysis result of the included studies that assessed the impact of each study on the overall magnitude of pneumonia among people living with HIV.**

the highest pooled prevalence was recorded between 2021 and 2023 at 14.50%, followed by the year between 2018 and 2020 at 13.4% and the lowest prevalence was recorded between 2013 and 2015 at 3.31%.The data from the pneumonia review suggests that the pooled prevalence has generally increased from 2013 to 2023. This increase in infection rates at the current time could be due to people giving less attention to these bacterial opportunistic infections.

## Strengths and limitations

A key strength of this systematic review and meta-analysis is that it is the first to determine the pooled prevalence estimates of opportunistic bacterial infections among people living with HIV in Ethiopia. However, this study may have certain drawbacks. Specifically, a small number of eligible papers were collected from the included regions, and no published data was available from the Afar, Gambela, Somali, and Benishangul-Gumuz regions. This exclusion may affect the overall national prevalence estimates of opportunistic bacterial infections in Ethiopia.

## Conclusions and recommendations

The pooled prevalence of tuberculosis and pneumonia among people living with HIV in Ethiopia is high, with an increasing trend observed from 2013 to 2023. This rising prevalence, regardless of its underlying cause, has the potential to negatively impact the health and well-being of people living with HIV. Consequently, policymakers and health planners should place a great deal of emphasis on implementing relevant prevention and control measures. Furthermore, extensive research is needed in this population across different regions of the country to fully understand the nature, dynamics, and risk factors associated with TB and pneumonia among people living with HIV.

## Supporting information

**S1 Table. PRISMA 2020 Checklist.** The checklists/protocols following during writing this systematic review and meta-analysis.
(DOCX)

**S2 Table. Search Strategy.** Comprehensive search strategy for common opportunistic bacterial infections among people living with HIV in Ethiopia.
(DOCX)

**S3 Table. Quality appraisal result of included studies.** The quality of included studies was evaluated using the Joanna Briggs Institute (JBI) quality appraisal checklist for prevalence studies.
(DOCX)

## Author contributions

**Conceptualization:** Abayeneh Girma, Aleka Aemiro, Getachew Alamnie.

**Data curation:** Aleka Aemiro, Getachew Alamnie, Demsew Beletew, Amere Genet.

**Formal analysis:** Abayeneh Girma, Demsew Beletew.

**Investigation:** Aleka Aemiro, Demsew Beletew, Amere Genet.

**Methodology:** Abayeneh Girma, Aleka Aemiro, Getachew Alamnie, Amere Genet.

**Project administration:** Abayeneh Girma.

**Resources:** Aleka Aemiro, Getachew Alamnie, Amere Genet.

**Software:** Abayeneh Girma, Demsew Beletew.

**Supervision:** Abayeneh Girma.

**Validation:** Abayeneh Girma, Aleka Aemiro, Getachew Alamnie, Demsew Beletew, Amere Genet.

**Visualization:** Abayeneh Girma, Aleka Aemiro, Getachew Alamnie, Demsew Beletew, Amere Genet.

**Writing – original draft:** Aleka Aemiro, Getachew Alamnie, Demsew Beletew, Amere Genet.

**Writing – review & editing:** Abayeneh Girma.

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
