## [Decision Letter · Decision Letter 0]

14 Aug 2024

We look forward to receiving your revised manuscript.

Kind regards,

Mengistu Hailemariam Zenebe, PhD

Academic Editor

PLOS ONE

Journal Requirements:

Reviewers' comments:

Reviewer's Responses to Questions

**Comments to the Author**

1. Is the manuscript technically sound, and do the data support the conclusions?

Reviewer #1: Partly

Reviewer #2: No

2. Has the statistical analysis been performed appropriately and rigorously?

Reviewer #1: Yes

Reviewer #2: Yes

3. Have the authors made all data underlying the findings in their manuscript fully available?

Reviewer #1: Yes

Reviewer #2: Yes

4. Is the manuscript presented in an intelligible fashion and written in standard English?

Reviewer #1: Yes

Reviewer #2: No

Reviewer #1: My comments and concerns are as follow:

Major

The focus of MS: Trends vs Systematic review & meta-analysis

The trends are not indicated properly.

What is the clear definition of pneumonia? Not clear and requires definition

If bacteriology confirmed pneumonia, the prevalence of predominant isolates should be reported.

How retrospective studies included in this study showed/confirmed the pneumonia?

Minor

Population: mainly focus on children, better to balance and align with objective of the study and title.

Is the study protocol registered in PRESPOROUS? If so, indicate the registration number

Language: better to use people/person first language. Thus, use ‘people with HIV’ than HIV positive patients.

I suggest the revisions of conclusion and recommendations according to the objective, title of the study and main findings

Good luck

Reviewer #2: COMMENTS

TITLE:

Prevalence and trends of opportunistic bacterial infections (Tuberculosis and pneumonia) among HIV positive patients in Ethiopia: Systematic review and meta-analysis

Comments: Since this study did not evaluate the trends of opportunistic infections (such as TB and pneumonia) among HIV-positive individuals, it would be inappropriate to include "trends of" in your title. Please remove "trend" from the title.

ABSTRACT

Conclusions: “This meta-analysis shows that the trend of tuberculosis infection is rising, then

falling, then rising again, while the trend of pneumonia infection among HIV-positive individuals

in Ethiopia is rising”.

Comments: If you’re certain that a trend analysis was conducted, please include the results in the abstract. However, I didn’t see this information in the results section.

Keywords: “ Ethiopia, meta-analysis, prevalence, pneumonia, tuberculosis, systemic review, and opportunistic infection”.

Comments: Better if you can make like

HIV(+)-people, opportunistic infections, tuberculosis, pneumonia, prevalence, systematic review, meta-analysis, Ethiopia

INTRODUCTION:

Comments: The introduction section does not adequately address Ethiopian studies on the prevalence of opportunistic infections, such as TB and pneumonia, among HIV-positive individuals in Ethiopia. Please revise the introduction to include this discussion.

AIDS

Comments: Please first provide the full definition of any term, followed by its abbreviation in parentheses, the first time it is mentioned in the document. After that, you may use the abbreviation consistently throughout the rest of the document.

“A 2019 report states that OI was the main cause of nearly 95,000 child deaths from

HIV-related causes [3].”

Comments: A 2019 global report…

“There were 310,000 OI-related fatalities among AIDS patients in the eastern and southern African regions, even though there is insufficient information about the OIs' recurrence rate in the continent [6].

Comment: please check the reference. "6. Teker AG. AIDS-related deaths in Turkey between 2009 and 2018. Epidemiology & Infection. 2021;149:e191."

“Globally, 9.9 million cases of tuberculosis and 214,000 HIV-positive deaths are expected in 2021, according to WHO estimates [9].”

Comments: Please review this sentence for accuracy, or consider using the most recent data.

“ Ethiopia is one of the 30 countries with a high TB and TB/HIV burden; with an estimated annual TB incidence of 140/100,000 persons and a death rate of 19 per 100,000 people, according to the 2020 Global TB Report [9]”.

Comments: Please refer to the updated WHO report on Ethiopia's TB burden, either the 2022 or 2023 edition.

METHODS

2. 1 Country profile

Comments: I believe the country profile information is not particularly relevant to this manuscript. Instead, it would be more appropriate for the authors to focus on the burden of HIV, TB, pneumonia, and other opportunistic infections in Ethiopia, and include related data that is pertinent to the manuscript.

If you believe the country profile is relevant to the manuscript, please revise it to include the most up-to-date information and provide appropriate citations or references.

2.2 Search strategy

….. ‘‘opportunistic bacterial infections in Ethiopia’’ ‘‘Prevalence of opportunistic bacterial

infections in HIV positive patients,’’ ‘‘Prevalence of opportunistic bacterial infections in Ethiopia,’’and ‘‘Magnitude of opportunistic bacterial infections among HIV positive patients in different regions of Ethiopia’’ were used to search journals.”

Comments: Please provide the search strategy you followed and the total number of articles captured in at least one database (e.g., PubMed) as a supplementary file. Ensure the information is formatted appropriately, either using "AND" or "OR" operators, or with commas, as needed.

……………..and study locations (conducted between September 14/2023 and December 1, 2023),

Comments: The above sentence is not clear, and needs revision

2.3 Inclusion and exclusion criteria of the studies

Comments: Please change with “eligibility criteria”

2.3.1 Inclusion criteria

Comments: This paragraph does not outline the inclusion criteria for potential papers in this systematic review and meta-analysis. Instead, it discusses the characteristics of the included studies. Please clearly state your inclusion criteria: specify which types of papers and necessary information were considered for inclusion, and detail which papers and types of information were deemed irrelevant or excluded.

2.3.2 Exclusion criteria

Comments: Please explicitly list all of your exclusion criteria.

What about papers that are only available as abstracts, conference presentations, or lack full-text access?

2.4. Study selection procedures

…. An EndNote X7 reference program,

Comments: Include the specifications for EndNote.

……inter-rater agreement was calculated after referring to the Cochrane Handbook of Systematic Reviews.

Comment: Cite the reference

2.7 Risk of publication bias

Comments: please remove this subheading and include the information (risk of publication bias into 2.8 Data analysis subheading section.

3. Results

Comments: include a subheading " Searching results"

“A total of 276 articles were retrieved on the prevalence and determinants of OIs among HIV positive patients were retrieved in Ethiopia”

Comments: Assessing the determinants of opportunistic infections among HIV-positive individuals is not the objective of your study. Please ensure that your focus remains on the actual objectives of your research.

“Of the remaining 27 articles, 7 articles were further removed for different purposes (lack of OR, CI and number of positive cases)”.

Comments: Why did you exclude articles that lacked OR and CI data, given that your primary outcome of interest is the prevalence of opportunistic infections (TB and pneumonia) among HIV-positive individuals? Since your study does not assess the determinants of these infections, it is unclear why these papers were excluded. Please revise this decision.

“Therefore, 20 of the studies met the eligibility criteria and were included in

the final systematic review and meta-analysis study (Figure 1)”.

Comments: Please revise the PRISMA flow chart as follows: Records after duplicates removed (n=151) → Records screened (n=151) → Full-text articles assessed for eligibility (n=27) → Studies included in the systematic review and meta-analysis (n=20).

3.1 Characteristics of the eligible studies

Comments: The paragraph requires significant editing and language revision.

Table 1: Prevalence of tuberculosis among HIV patients in Ethiopia

Comments: Characters-tics of included studies to calculate the pooled prevalence of TB among HIV(+)-individuals

Table 2: Prevalence of pneumonia among HIV patients in Ethiopia

Comments: Characteristics of included studies to calculate the pooled prevalence of pneumonia among HIV(+)-individuals

3.2 Pooled prevalence of opportunistic bacterial infections

Comments: Pooled prevalence of tuberculosis among HIV(+) individuals

4. Discussion

Comments:

The entire document requires thorough language editing, including improvements in sentence structure, grammar, and overall clarity. Careful attention should be given to revising the text to ensure it is well-constructed, grammatically correct, and easy to understand.

Your study presents a pooled or combined prevalence, which involves aggregating data from multiple sources to provide a more comprehensive and reliable estimate. Given the nature of this approach, it may not be appropriate or recommended to compare your findings with individual primary studies, as these studies typically offer a narrower scope and may not capture the broader trends that your meta-analysis addresses. Instead, it would be more suitable and meaningful to discuss your results in the context of other meta-analyses or national reports. These sources provide a wider perspective and are more comparable to your pooled data, allowing for a more accurate and robust discussion of your findings. By focusing on these broader sources, you can better validate your results and place them within the larger body of evidence.

5.1 Conclusions

“In terms of the prevalence of tuberculosis, the pattern is rising, falling, and rising.

However, pneumonia prevalence among HIV positive patients is increasing”.

Comments: You did not estimate the pooled prevalence of TB across different time intervals, so this statement is not accurate. Please ensure that your conclusions are precise and based directly on the findings of your study.

**Do you want your identity to be public for this peer review?** For information about this choice, including consent withdrawal, please see our Privacy Policy

Reviewer #1: No

Reviewer #2: **Yes: ** Melese Abate Reta

---

## [Author Response · Author response to Decision Letter 1]

4 Oct 2024

Dear reviewers, we are grateful for your time and effort, and all your comments and suggestions are constructive and supportive, which increases the readability and overall quality of the paper. Below, we provide a point-by-point response explaining how we have addressed each of the reviewer's comments and suggestions. N.B., yellow and bright green, highlighted, are the answers for reviewers 1 and 2, respectively, and turquoise color for both reviewers. We hope that the revised manuscript will better fit the journal.

General Comments to the Author

Recommendation 1: Is the manuscript technically sound, and do the data support the conclusions?

Reviewer #1: Partly

Reviewer #2: No

Answer 1: Dear reviewers, Thanks for your valuable recommendation; we accepted it and corrected as per your suggestion.

Recommendation 2: Has the statistical analysis been performed appropriately and rigorously?

Reviewer #1: Yes

Reviewer #2: Yes

Answer 2: Dear reviewers, Thanks for your appreciation.

Recommendation 3: Have the authors made all data underlying the findings in their manuscript fully available?

Reviewer #1: Yes

Reviewer #2: Yes

Answer 3: Dear reviewers, Thanks for your appreciation.

Recommendation 4: Is the manuscript presented in an intelligible fashion and written in standard English?

Reviewer #1: Yes

Reviewer #2: No

Answer 4: Dear reviewer 2, Thanks for your valuable recommendation; we accepted it and corrected as per your suggestion.

Answer for Reviewer 1#

Major

Recommendation 1: The focus of MS: Trends vs Systematic review & meta-analysis. The trends are not indicated properly.

Answer 1: Dear reviewer, Thanks for your valuable recommendation, but the trend is not relevant and was removed.

Recommendation 2: What is the clear definition of pneumonia? Not clear and requires definition.

Answer 2: Dear reviewer, Thanks for your valuable recommendation; we accepted it and corrected as per your suggestion. Could you please go to the introduction?

Recommendation 3: If bacteriology confirmed pneumonia, the prevalence of predominant isolates should be reported.

Answer 3: Dear reviewer, Thanks for your valuable recommendation, but they are not mentioned in the primary studies.

Recommendation 4: How retrospective studies included in this study showed/confirmed the pneumonia?

Answer 4: Dear reviewer, Thanks for your valuable recommendation; but it was and editorial problem and now amended. Could you please go to the result?

Minor

Recommendation 30: Population: mainly focus on children, better to balance and align with objective of the study and title.

Answer 30: Dear reviewer, Thanks for your valuable recommendation; we accepted it and corrected as per your suggestion. Could you please go to the introduction?

Recommendation 31: Is the study protocol registered in PRESPOROUS? If so, indicate the registration number.

Answer 31: Dear reviewer, yes it is already registered and found in abstract methodology section.

Recommendation 32: Language: better to use people/person first language. Thus, use ‘people with HIV’ than HIV positive patients.

Answer 32: Dear reviewer, Thanks for your valuable recommendation; we accepted it and corrected as per your suggestion. Could you please go to the entire document of the manuscript?

Recommendation 33: I suggest the revisions of conclusion and recommendations according to the objective and title of the study and main findings.

Answer 33: Dear reviewer, Thanks for your valuable recommendation; we accepted it and corrected as per your suggestion. Could you please go to the conclusion and recommendation?

Answer for Reviewer 2#

Recommendation 1: Since this study did not evaluate the trends of opportunistic infections (such as TB and pneumonia) among HIV-positive individuals, it would be inappropriate to include "trends of" in your title. Please remove "trend" from the title.

Answer 1: Dear reviewer, Thanks for your valuable recommendation; we accepted it and corrected as per your suggestion. Could you please go to the title and other parts of the manuscript?

Recommendation 2: If you’re certain that a trend analysis was conducted, please include the results in the abstract. However, I didn’t see this information in the results section.

Answer 2: Dear reviewer, Thanks for your valuable recommendation; we accepted it and corrected as per your suggestion. Could you please go to the abstract?

Recommendation 3: Better if you can make the key words like, HIV (+)-people, opportunistic infections, tuberculosis, pneumonia, prevalence, systematic review, meta-analysis, Ethiopia

Answer 3: Dear reviewer, Thanks for your valuable recommendation; we accepted it and corrected as per your suggestion. Could you please go to the abstract?

Introduction:

Recommendation 4: The introduction section does not adequately address Ethiopian studies on the prevalence of opportunistic infections, such as TB and pneumonia, among HIV-positive individuals in Ethiopia. Please revise the introduction to include this discussion.

Answer 4: Dear reviewer, Thanks for your valuable recommendation; we accepted it and corrected as per your suggestion. Could you please go to the introduction?

Recommendation 5: Please first provide the full definition of any term, followed by its abbreviation in parentheses, the first time it is mentioned in the document. After that, you may use the abbreviation consistently throughout the rest of the document. “A 2019 report states that OI was the main cause of nearly 95,000 child deaths from HIV-related causes [3].”

Answer 5: Dear reviewer, Thanks for your valuable recommendation; we accepted it and corrected as per your suggestion. Could you please go to the introduction?

Recommendation 6: A 2019 global report… “There were 310,000 OI-related fatalities among AIDS patients in the eastern and southern African regions, even though there is insufficient information about the OIs' recurrence rate in the continent [6].

Answer 6: Dear reviewer, Thanks for your valuable recommendation; we accepted it and corrected as per your suggestion. Could you please go to the introduction?

Recommendation 7: please check the reference. "6. Teker AG. AIDS-related deaths in Turkey between 2009 and 2018. Epidemiology & Infection. 2021;149:e191." “Globally, 9.9 million cases of tuberculosis and 214,000 HIV-positive deaths are expected in 2021, according to WHO estimates [9].”

Answer 7: Dear reviewer, Thanks for your valuable recommendation; we accepted it and corrected as per your suggestion. Could you please go to the introduction?

Recommendation 8: Please review this sentence for accuracy, or consider using the most recent data. “Ethiopia is one of the 30 countries with a high TB and TB/HIV burden; with an estimated annual TB incidence of 140/100,000 persons and a death rate of 19 per 100,000 people, according to the 2020 Global TB Report [9]”.

Recommendation 9: Please refer to the updated WHO report on Ethiopia's TB burden, either the 2022 or 2023 edition

Answer 9: Dear reviewer, Thanks for your valuable recommendation; we accepted it and corrected as per your suggestion. Could you please go to the introduction?

Methods

Recommendation 10: Country profile: I believe the country profile information is not particularly relevant to this manuscript. Instead, it would be more appropriate for the authors to focus on the burden of HIV, TB, pneumonia, and other opportunistic infections in Ethiopia, and include related data that is pertinent to the manuscript. If you believe the country profile is relevant to the manuscript, please revise it to include the most up to-date information and provide appropriate citations or references.

Answer 10: Dear reviewer, Thanks for your valuable recommendation; we accepted it and corrected as per your suggestion. Could you please go to the methodology?

Recommendation 11: Please provide the search strategy you followed and the total number of articles captured in at least one database (e.g., PubMed) as a supplementary file. Ensure the information is formatted appropriately, either using "AND" or "OR" operators, or with commas, as needed.

Answer 11: Dear reviewer, Thanks for your valuable recommendation; we accepted it and corrected as per your suggestion. Could you please go to the methodology?

Recommendation 12: ……………..and study locations (conducted between September 14/2023 and December 1, 2023). The above sentence is not clear, and needs revision.

Answer 12: Dear reviewer, Thanks for your valuable recommendation; we accepted it and corrected as per your suggestion. Could you please go to the methodology?

Recommendation 13: Please change inclusion and exclusion criteria of the studies with “eligibility criteria”

Answer 13: Dear reviewer, Thanks for your valuable recommendation; we accepted it and corrected as per your suggestion. Could you please go to the methodology?

Recommendation 14: An Inclusion criterion, this paragraph does not outline the inclusion criteria for potential papers in this systematic review and meta-analysis. Instead, it discusses the characteristics of the included studies. Please clearly state your inclusion criteria: specify which types of papers and necessary information were considered for inclusion, and detail which papers and types of information were deemed irrelevant or excluded.

Answer 14: Dear reviewer, Thanks for your valuable recommendation; we accepted it and corrected as per your suggestion. Could you please go to the methodology?

Recommendation 15: Exclusion criteria, please explicitly list all of your exclusion criteria. What about papers that are only available as abstracts, conference presentations, or lack full-text access?

Answer 15: Dear reviewer, Thanks for your valuable recommendation; we accepted it and corrected as per your suggestion. Could you please go to the methodology?

Recommendation 16: Study selection procedures …. An EndNote X7 reference program, Include the specifications for EndNote. ……inter-rater agreement was calculated after referring to the Cochrane Handbook of Systematic Reviews. Cite the reference

Answer 16: Dear reviewer, Thanks for your valuable recommendation; we accepted it and corrected as per your suggestion. Could you please go to the methodology?

Recommendation 17: Risk of publication bias please removes this subheading and includes the information (risk of publication bias into 2.8 Data analysis subheading section.

Answer 17: Dear reviewer, Thanks for your valuable recommendation; we accepted it and corrected as per your suggestion. Could you please go to the methodology?

Results

Recommendation 18: include a subheading "Searching results" “A total of 276 articles were retrieved on the prevalence and determinants of OIs among HIV positive patients were retrieved in Ethiopia.

Answer 18: Dear reviewer, Thanks for your valuable recommendation; we accepted it and corrected as per your suggestion. Could you please go to the result?

Recommendation 19: Assessing the determinants of opportunistic infections among HIV-positive individuals is not the objective of your study. Please ensure that your focus remains on the actual objectives of your research. “Of the remaining 27 articles, 7 articles were further removed for different purposes (lack of OR, CI and number of positive cases)”.

Answer 19: Dear reviewer, Thanks for your valuable recommendation; we accepted it and corrected as per your suggestion. Could you please go to the result?

Recommendation 20: Why did you exclude articles that lacked OR and CI data, given that your primary outcome of interest is the prevalence of opportunistic infections (TB and pneumonia) among HIV- positive individuals? Since your study does not assess the determinants of these infections, it is unclear why these papers were excluded. Please revise this decision. “Therefore, 20 of the studies met the eligibility criteria and were included in the final systematic review and meta-analysis study (Figure 1)”.

Answer 20: Dear reviewer, Thanks for your valuable recommendation; we accepted it and corrected as per your suggestion. Could you please go to the result?

Recommendation 21: Please revise the PRISMA flow chart as follows: Records after duplicates removed (n=151) → Records screened (n=151) → Full-text articles assessed for eligibility (n=27) → Studies included in the systematic review and meta-analysis (n=20).

Answer 21: Dear reviewer, Thanks for your valuable recommendation; we accepted it and corrected as per your suggestion. Could you please go to the result?

Characteristics of the eligible studies

Recommendation 22: The paragraph requires significant editing and language revision

Answer 22: Dear reviewer, Thanks for your valuable recommendation; we accepted it and corrected as per your suggestion. Could you please go to the result?

Recommendation 23: Characteristics of included studies to calculate the pooled prevalence of TB among HIV (+) individuals on the prevalence of tuberculosis among HIV patients in Ethiopia

Answer 23: Dear reviewer, Thanks for your valuable recommendation; we accepted it and corrected as per your suggestion. Could you please go to the result?

Recommendation 24: Characteristics of included studies to calculate the pooled prevalence of pneumonia among HIV (+)-individuals on prevalence of pneumonia among HIV patients in Ethiopia

Answer 24: Dear reviewer, Thanks for your valuable recommendation; we accepted it and corrected as per your suggestion. Could you please go to the result?

Recommendation 25: Correct pooled prevalence of opportunistic bacterial infections to pooled prevalence of tuberculosis among HIV (+) individuals.

Answer 25: Dear reviewer, Thanks for your valuable recommendation; we accepted it and corrected as per your suggestion. Could you please go to the result?

Discussion

Recommendation 26: The entire document requires thorough language editing, including improvements in sentence structure, grammar, and overall clarity. Careful attention should be given to revising the text to ensure it is well-constructed, grammatically correct, and easy to understand. Your study presents a pooled or combined prevalence, which involves aggregating data from multiple sources to provide a more comprehensive and reliable estimate. Given the nature of this approach, it may not be appropriate or recommended to compare your findings with individual primary studies, as these studies typically offer a narrower scope and may not capture the broader trends that your meta-analysis addresses. Instead, it would be more suitable and meaningful to discuss your results in the context of other meta-analyses or national reports. These sources provide a wider perspective and are more comparable to your pooled data, allowing for a more accurate and robust discussion of your findings. By focusing on these broader sources, you can better validate your results and place them within the larger body of evidence.

Answer 26: Dear reviewer, Thanks for your valuable recommendation; we accepted it and corrected as per your suggestion. Could you please go to the Discussion?

Conclusions

Recomm

---

## [Decision Letter · Decision Letter 1]

14 Nov 2024

Dear Dr. Girma,

Thank you for submitting your manuscript to PLOS ONE. After careful consideration, we feel that it has merit but does not fully meet PLOS ONE’s publication criteria as it currently stands. Therefore, we invite you to submit a revised version of the manuscript that addresses the points raised during the review process.

We look forward to receiving your revised manuscript.

Kind regards,

Mengistu Hailemariam Zenebe, PhD

Academic Editor

PLOS ONE

Journal Requirements:

Reviewers' comments:

Reviewer's Responses to Questions

**Comments to the Author**

Reviewer #2: All comments have been addressed

Reviewer #3: (No Response)

2. Is the manuscript technically sound, and do the data support the conclusions?

Reviewer #2: Yes

Reviewer #3: Yes

3. Has the statistical analysis been performed appropriately and rigorously?

Reviewer #2: Yes

Reviewer #3: I Don't Know

4. Have the authors made all data underlying the findings in their manuscript fully available?

Reviewer #2: (No Response)

Reviewer #3: Yes

5. Is the manuscript presented in an intelligible fashion and written in standard English?

Reviewer #2: Yes

Reviewer #3: No

Reviewer #2: (No Response)

Reviewer #3: Firstly, I would like to thank the editor for inviting me to review this manuscript entitled prevalence of opportunistic bacterial infections (Tuberculosis and Pneumonia) among people living with HIV in Ethiopia: A systematic review and meta-analysis.

The authors’ relentless effort in conducting this interesting area of research is also appreciated. Knowing the burden of the problem can help plan strategies for responsible officials in the future. Saying this, I have some comments that need to be addressed before its consideration for publication.

At the very beginning, there is a study conducted previously with the same title and study area and even broader than the current study (prevalence and determinants of opportunistic infections among HIV-infected adults receiving ART in Ethiopia: A systematic review and meta-analysis). If so what is the main reason to conduct this study at the moment for fear of duplication effort?

Introduction

1. The introduction lacks chronological order; it would be better if the authors were encouraged to revise it as the following:

-Define the problem clearly, and report the burden of the problem from global to national or from developed to developing nations

-Explain the impacts/consequences of the problem if left unaddressed or untreated

-The possible risk factors of opportunistic infections in people with HIV (of course this is not your objective you can skip)

-Finally mention what has been done previously to tackle the problem and gaps your study is going to address. Here boldly show the justification of the current study.

2. Avoid citing wrong references. For instance, refer. 6

3. Focus on your objective only (pneumonia among non-RVI children is not objective so avoid it) and the introduction is mostly about pneumonia ignoring tuberculosis why?

Methods

1. In search strategy Google Scholar is not a database remove it.

2. Merge the inclusion and exclusion criteria into eligibility criteria and almost all articles from Ethiopia published in English. So this shouldn’t be a criteria remove it

3. Quality appraisal put cut off point of JBI for studies to be included or excluded

4. Data analysis: what is your base to conclude I2 25% is low and 75% high heterogeneity or p<0.05 significant publication bias? Put citations

Results

1. Bring the study selection procedure under the result subsection

2. This has publication so what did you do for that?

3. To identify the possible source of heterogeneity please consider meta-regression

4. When conducting subgroup analysis please try to merge smaller numbers into one as others (eg. Tigray region can be merged with Oromia or Amhara in the TB subgroup)

5. In the figures of sensitivity analysis (fig. 6) the estimates should be the same with pooled prevalence of TB and pneumonia.

Conclusion and recommendations

1. Try to conclude based on your study’s findings not based on your general knowledge.

2. In the recommendation section you mentioned that “Organize regular training sessions for healthcare workers to enhance their knowledge regarding opportunistic infections, diagnostic criteria, and treatment protocols specific to HIV-positive patients. Provide healthcare professionals with updated clinical guidelines, treatment algorithms, and educational resources that focus on the management of opportunistic infections in the context of HIV.” The question does your study reveal that the increment of opportunistic infection among people living with HIV is due to a lack of trained professionals or poor knowledge?

Generally, intense language editing is needed throughout the whole manuscript to make it clearer and easily understandable for readers. Please avoid using expressions like “HIV-positive” or “HIV patients” throughout the whole document they seem to be stigmatizing instead replace them with “people living with HIV” or “people with HIV”.

**Do you want your identity to be public for this peer review?** For information about this choice, including consent withdrawal, please see our Privacy Policy

Reviewer #2: **Yes: ** Melese Abate Reta

Reviewer #3: No

---

## [Author Response · Author response to Decision Letter 2]

19 Nov 2024

Mengistu Hailemariam Zenebe, PhD

Academic Editor

PLOS ONE

November 19, 2024

Dear Dr. Mengistu Hailemariam Zenebe,

We are pleased to submit the revised draft of our manuscript, “Prevalence of opportunistic bacterial infections (tuberculosis and pneumonia) among people with HIV in Ethiopia: Systematic review and meta-analysis” (PONE-D-24-22113R1), to PLOS ONE. We appreciate the time and effort dedicated by the editorial staff and reviewers. The comments provided were valuable and helped us refine our paper. As such, we have made several revisions to the manuscript based on the suggestions given. Changes to the manuscript are yellow highlighted.

Below are our point-by-point responses to the reviewers’ comments.

Response to Reviewer 3

Thank you for your insightful comments and suggestions. Please find the answers to each of your questions below.

1. At the very beginning, there is a study conducted previously with the same title and study area and even broader than the current study (prevalence and determinants of opportunistic infections among HIV-infected adults receiving ART in Ethiopia: A systematic review and meta-analysis). If so what is the main reason to conduct this study at the moment for fear of duplication effort?

Response: Previously, Woldegeorgis and colleagues [14] conducted a systematic review and meta-analysis regarding the prevalence and determinants of opportunistic infections among HIV-infected adults receiving ART in Ethiopia. However, the prevalence of opportunistic bacterial infections, particularly tuberculosis and pneumonia, among people of all age groups living with HIV in the country is not collected, well organised, and documented as a systematic review and meta-analysis. Therefore, the objective of this systematic review and meta-analysis is to estimate the overall prevalence of opportunistic bacterial infections among people living with HIV from available research conducted in different regions of Ethiopia. The findings of this work will be used by the concerned stakeholders to reduce the prevalence of opportunistic bacterial infections and design evidence-based interventions (page 4, last paragraph).

2. Introduction

1. The introduction lacks chronological order; it would be better if the authors were encouraged to revise it as the following:

-Define the problem clearly, and report the burden of the problem from global to national or from developed to developing nations

-Explain the impacts/consequences of the problem if left unaddressed or untreated

-The possible risk factors of opportunistic infections in people with HIV (of course this is not your objective you can skip)

-Finally mention what has been done previously to tackle the problem and gaps your study is going to address. Here boldly show the justification of the current study.

2. Avoid citing wrong references. For instance, refer. 6

3. Focus on your objective only (pneumonia among non-RVI children is not objective so avoid it) and the introduction is mostly about pneumonia ignoring tuberculosis why?

Response: Thank you for your close reading of our paper. This is an important point, so we have amended as per your suggestions (pages 3 & 4).

3. Methods

1. In search strategy Google Scholar is not a database remove it.

2. Merge the inclusion and exclusion criteria into eligibility criteria and almost all articles from Ethiopia published in English. So this shouldn’t be a criteria remove it

3. Quality appraisal put cut off point of JBI for studies to be included or excluded

4. Data analysis: what is your base to conclude I2 25% is low and 75% high heterogeneity or p<0.05 significant publication bias? Put citations

Response: We agree with your suggestion and have modified accordingly (pages 5-8).

4. Results

1. Bring the study selection procedure under the result subsection

2. This has publication so what did you do for that?

3. To identify the possible source of heterogeneity please consider meta-regression

4. When conducting subgroup analysis please try to merge smaller numbers into one as others (eg. Tigray region can be merged with Oromia or Amhara in the TB subgroup)

5. In the figures of sensitivity analysis (fig. 6) the estimates should be the same with pooled prevalence of TB and pneumonia.

Response: Thank you for this observation our answer for all of your suggestions are found below. This is because (1) study selection process must be included in the method section as per “PRISMA-2020 checklist”. Please check the “Supplemental File 1”. The second (2) comment is not clear. (3) Dear reviewer, regarding identifying the possible source of heterogeneity using meta-regression is optional and our team chooses the figure representation rather than the meta-regression. (4) We did not do it because Tigray is one of the regions found in Ethiopia and merging with Oromia or Amhara region results falsified report or due to this study, stakeholders may not consider Tigray region particularly during reducing the prevalence of opportunistic bacterial infections and design evidence-based interventions. With regard to sensitivity analysis (5) As far as we know, a sensitivity analysis is carried out by removing each study one at a time in order to clarify the impact of each study on the pooled effect size. During the sensitivity analysis, studies not included in Figs 6a & b had relatively determinant effects on the overall magnitude of tuberculosis and pneumonia among people living with HIV in Ethiopia, so we removed them and this is scientifically valid way of reporting the sensitivity analysis. Dear reviewer, in this regard we recommended to see more information from different previously published works elsewhere or get in touch with statistician.

5. Conclusion and recommendations

1. Try to conclude based on your study’s findings not based on your general knowledge.

2. In the recommendation section you mentioned that “Organize regular training sessions for healthcare workers to enhance their knowledge regarding opportunistic infections, diagnostic criteria, and treatment protocols specific to HIV-positive patients. Provide healthcare professionals with updated clinical guidelines, treatment algorithms, and educational resources that focus on the management of opportunistic infections in the context of HIV.” The question does your study reveal that the increment of opportunistic infection among people living with HIV is due to a lack of trained professionals or poor knowledge?

Response: We agree with your suggestion and have modified accordingly (pages 18 & 19).

6. Generally, intense language editing is needed throughout the whole manuscript to make it clearer and easily understandable for readers. Please avoid using expressions like “HIV-positive” or “HIV patients” throughout the whole document they seem to be stigmatizing instead replace them with “people living with HIV” or “people with HIV”.

Response: We agree with your suggestion and have modified accordingly (throughout the manuscript).

Sincerely,

Abayeneh Girma

Corresponding author

---

## [Decision Letter · Decision Letter 2]

18 Dec 2024

PONE-D-24-22113R2

Prevalence of opportunistic bacterial infections (tuberculosis and pneumonia) among people with HIV in Ethiopia: Systematic review and meta-analysis

PLOS ONE

Dear Dr. Girma,

Thank you for submitting your manuscript to  PLOS ONE., and for responding to our recent requests regarding your submission. Unfortunately, in our final editorial checks of the documents that you supplied, we have concluded that your submission does not comply with our policies around data availability. We are therefore overturning the provisional editorial accept decision, and rejecting this manuscript.  

PLOS journals require authors to make all data necessary to replicate their study’s findings publicly available without restriction at the time of publication (https://journals.plos.org/plosone/s/data-availability). In this case, the following underlying data were not provided as repeatedly requested: 

A table of all data extracted from the primary research sources for the systematic review and/or meta-analysis.  

As a result of these concerns, we cannot consider the manuscript for publication. I am very sorry that this issue was identified at such a late stage.  

Kind regards,

Vanessa Carels

Staff Editor

PLOS ONE

Reviewers' comments:

Reviewer's Responses to Questions

**Comments to the Author**

Reviewer #2: All comments have been addressed

Reviewer #3: All comments have been addressed

2. Is the manuscript technically sound, and do the data support the conclusions?

Reviewer #2: Yes

Reviewer #3: Yes

3. Has the statistical analysis been performed appropriately and rigorously?

Reviewer #2: Yes

Reviewer #3: Yes

4. Have the authors made all data underlying the findings in their manuscript fully available?

Reviewer #2: Yes

Reviewer #3: Yes

5. Is the manuscript presented in an intelligible fashion and written in standard English?

Reviewer #2: Yes

Reviewer #3: Yes

Reviewer #2: (No Response)

Reviewer #3: (No Response)

**Do you want your identity to be public for this peer review?** For information about this choice, including consent withdrawal, please see our Privacy Policy

Reviewer #2: **Yes: ** Melese Abate Reta

Reviewer #3: No

- - - - -

---

## [Author Response · Author response to Decision Letter 3]

28 Dec 2024

There is nothing to respond to from the decision letter since that was an acceptance letter.

---

## [Decision Letter · Decision Letter 3]

22 Sep 2025

Prevalence of opportunistic bacterial infections (tuberculosis and pneumonia) among people with HIV in Ethiopia: Systematic review and meta-analysis

PONE-D-24-22113R3

Dear Dr. Girma,

We’re pleased to inform you that your manuscript has been judged scientifically suitable for publication and will be formally accepted for publication once it meets all outstanding technical requirements.

Kind regards,

Miquel Vall-llosera Camps

Senior Staff Editor

PLOS One

Reviewers' comments:

Reviewer's Responses to Questions

**Comments to the Author**

Reviewer #3: All comments have been addressed

2. Is the manuscript technically sound, and do the data support the conclusions?

Reviewer #3: Yes

3. Has the statistical analysis been performed appropriately and rigorously?

Reviewer #3: Yes

4. Have the authors made all data underlying the findings in their manuscript fully available?

Reviewer #3: Yes

5. Is the manuscript presented in an intelligible fashion and written in standard English?

Reviewer #3: Yes

Reviewer #3: (No Response)

**Do you want your identity to be public for this peer review?** For information about this choice, including consent withdrawal, please see our Privacy Policy

Reviewer #3: No

---

## [Editor Report · Acceptance letter]

PONE-D-24-22113R3

PLOS ONE

Dear Dr. Girma,

I'm pleased to inform you that your manuscript has been deemed suitable for publication in PLOS ONE. Congratulations! Your manuscript is now being handed over to our production team.

Kind regards,

on behalf of

Dr. Miquel Vall-llosera Camps

Staff Editor

PLOS ONE